# Relationships between Occupational Stress, Change in Work Environment during the COVID-19 Pandemic, and Depressive and Anxiety Symptoms among Non-Healthcare Workers in Japan: A Cross-Sectional Study

**DOI:** 10.3390/ijerph19020983

**Published:** 2022-01-16

**Authors:** Yasuhiko Deguchi, Shinichi Iwasaki, Akihiro Niki, Aya Kadowaki, Tomoyuki Hirota, Yoshiki Shirahama, Yoko Nakamichi, Yutaro Okawa, Yuki Uesaka, Koki Inoue

**Affiliations:** Department of Neuropsychiatry, Graduate School of Medicine, Osaka City University, Osaka 545-8585, Japan; siwasaki@med.osaka-cu.ac.jp (S.I.); aki29seiya@yahoo.co.jp (A.N.); ak.peridot@gmail.com (A.K.); tomo19890323@gmail.com (T.H.); skybluegreen.2112@gmail.com (Y.S.); yoko.nakamichi.4@gmail.com (Y.N.); koro3koro3jp@gmail.com (Y.O.); uesakayuki@yahoo.co.jp (Y.U.); kokii@med.osaka-cu.ac.jp (K.I.)

**Keywords:** COVID-19, occupational stress, mental health, non-healthcare workers, job future ambiguity, variance in workload, depressive symptoms, anxiety symptoms

## Abstract

This study aims to clarify the effect of occupational stress and changes in the work environment on non-healthcare workers’ (HCWs) mental health during the third wave of the COVID-19 pandemic in Japan. A web-based, cross-sectional survey was conducted from 16 to 17 December 2020. Data from 807 non-HCWs were included. We evaluated occupational stress using the Generic Job Stress Questionnaire (GJSQ). Depressive and anxiety symptoms were assessed using the Japanese version of the Patient Health Questionnaire-9 and the Generalized Anxiety Disorder 7-item scale, respectively. We collected demographic variables, work-related variables, and the variables associated with COVID-19. The adjusted odds ratios for depressive and anxiety groups were estimated using multivariate logistic regression analyses, adjusted for all the demographic variables, work-related variables, COVID-19-related variables, and the six subdivided GJSQ subscales. The results confirm a relationship between variance in workload, job future ambiguity, social support from coworkers, having contact with COVID-19 patients, and depressive and anxiety symptoms. Paying attention to job future ambiguity, the variance in workload at the workplace and individual perspectives, promoting contact and support among coworkers using online communication tools, and reducing contact with COVID-19 patients, will be useful for decreasing the depressive and anxiety symptoms among non-HCWs.

## 1. Introduction

Since 2020, the coronavirus disease 2019 (COVID-19) pandemic has been the main concern worldwide. As of 14 November 2021, over 252 million confirmed cases and more than 5 million deaths were reported by the World Health Organization [1]. In Japan, states of emergency were declared four times—March to May 2020, January to March, April to June, and July to September 2021—and efforts to maintain physical distancing and self-isolation were enforced. The prolonged COVID-19 pandemic also had a significant impact on global mental health.

The pandemic and the related containment measures (quarantine, physical distancing, and self-isolation) can have a detrimental impact on mental health. Concerns about one’s health and that of their loved ones, as well as the uncertainty about the future, can generate or exacerbate fear, depression, and anxiety, and these concerns, if prolonged, can increase the risk of serious and disabling mental health conditions among adult males and females [2]. A longitudinal study in the UK demonstrated prolonged deterioration in mental health for all age groups and genders, not only immediately after the initial lockdown, but also in the subsequent months when restrictions were eased [3]. Fiorillo and Gorwood stated that the pandemic will be over, but its effects on the mental health and well-being of the general population, health professionals, and vulnerable people will remain for a long time [2]. Therefore, further research that aims to evaluate the impact of the pandemic on mental health is needed.

Among the general global population, relatively high rates of depressive symptoms, from 14.6% to 48.3% [4,5,6], and anxiety symptoms, from 6.33% to 50.9% [4,5,6], have been reported during the COVID-19 pandemic in three systematic reviews and a multinational meta-analysis. In addition, these studies identified several risk factors associated with depressive and anxiety symptoms [4,5,6]: being female, in a younger age group, a student, having a lower socioeconomic status (e.g., living in rural areas, having an unstable income, and having lower education), and being unemployed; experiencing loneliness; being divorced, widowed, or single; not having a child; worrying about being infected; being at a high risk of contracting COVID-19; and frequent exposure to social media or news about COVID-19.

The COVID-19 pandemic overwhelmed hospitals worldwide, and healthcare workers (HCWs) have faced risks of poor mental health conditions. Systematic reviews and meta-analyses have consistently demonstrated an increased incidence of poor mental health, including elevated depressive and anxiety symptoms, psychological burden, and stress reaction, among HCWs during the COVID-19 pandemic [5,7,8,9,10,11,12,13,14,15]. In addition, previous research found differences in the mental health conditions between HCWs and non-HCWs in the early phase of the COVID-19 pandemic [16,17,18]. For example, two studies in China reported that HCWs and non-HCWs experienced similar levels of anxiety and depression [16], and the prevalence of depression and anxiety was greater among HCWs than non-HCWs [17]. Another study conducted during the first wave of the COVID-19 pandemic in Japan demonstrated that psychological distress, including fatigue, anxiety, and depression, increased significantly more among HCWs than non-HCWs [18].

However, the global COVID-19 pandemic has continued for more than a year, and its psychological effects on non-HCWs may have further increased. In fact, the work environment dramatically changed during the pandemic. For instance, the prevention of infection, using methods that include avoiding situations with the “three Cs” (i.e., enclosed space, crowding, and closed contact), physical distancing among coworkers or clients, and remote working, became a requirement. Further, the anxiety and fear regarding workplace infection, future income or workplace prospects, and unemployment continue to affect non-HCWs. A study including 123,768 factory workers in China, reported that the prevalence of depressive and anxiety symptoms was 22.8% and 3.4%, and having COVID-19 confirmed cases in the community, having COVID-19 confirmed friends, a poor health status, and alcoholism were associated with an increased risk of depressive/anxiety symptoms [19]. However, few studies examined the psychological effect of the COVID-19 pandemic on non-HCWs.

Before the COVID-19 pandemic, longitudinal studies and meta-analyses demonstrated that having a higher quantitative workload, lower job control, lower social support, and higher job strain was associated with an elevated risk of depressive symptoms [20,21,22,23]. Workers exposed to high psychological demands and low job control have a higher tendency to take sickness absence due to a mental disorder than workers with no exposure to such demands [24]. Some studies on occupational stress among HCWs have been conducted during the COVID-19 pandemic. A study including neurologists in Norway, reported that changed work routines and access to resources and the perception that medical follow-ups were unsatisfactory were associated with a high degree of burden and stress; however, the fear of becoming infected and ill was not an important contributor [25]. Another study with frontline nurses in China reported that the number of daily working hours, service years, and weekly night shifts, and the level of academic qualification were major factors related to job stress [26]. 

However, little is known about (a) what occupational stress factors impact non- HCWs’ mental health during the COVID-19 pandemic and (b) what work environment factor changes due to COVID-19 impact non-HCWs’ mental health. Therefore, we hypothesized that occupational stress during the COVID-19 pandemic and work environment changes due to the pandemic would significantly impact non-HCWs’ mental health. This study aims to clarify the effect of occupational stress and change in the work environment on non-HCWs’ mental health during the third wave of the COVID-19 pandemic in Japan.

## 2. Materials and Methods

### 2.1. Study Design, Participants, and Procedure

A web-based, cross-sectional survey was conducted in Japan through an online research company, Macromill, Inc. Japan, from 16 to 17 December 2020, which was during the third wave of the COVID-19 pandemic. The study used the following inclusion criteria for participants: (a) living in Japan, (b) being employed, and (c) being between 20 and 65 years of age. Previous studies using the Patient Health Questionnaire-9 (PHQ-9) and the Generalized Anxiety Disorder 7-item scale (GAD-7) reported prevalence rates of depressive and anxiety symptoms in the general population to be about 30–40% [27,28,29]. We aimed to recruit about 1000 Japanese workers with different employment statuses from a pool of approximately 10 million individuals registered with Macromill, Inc. A total of 1070 workers participated. We excluded participants with at least one missing entry on the questionnaire and 58 individuals who described themselves as HCWs in order to have a sample of only non-HCWs. Thus, non-HCWs consisted of all kinds of workers except HCWs, and the final analytic sample included a total of 807 eligible participants. 

### 2.2. Measures of Occupational Stress

We evaluated occupational stress using the Generic Job Stress Questionnaire (GJSQ) developed by the National Institute for Occupational Safety and Health (NIOSH) [30]. The Japanese version of the GJSQ demonstrated sufficient reliability and validity [31,32]. The NIOSH permits that the GJSQ subscales be used independently to evaluate occupational stress [30]. Based on the NIOSH job stress model [30], we focused on four subscales (quantitative workload, job control, variance in workload, and job future ambiguity) to evaluate occupational stress. Two subscales were used to evaluate social support (from supervisors and coworkers), which functions as a buffering factor, according to the results of many previous studies on the relationship between psychiatric symptoms and occupational stress [20,21,22,23,33,34,35,36,37]. Items on the GJSQ are positively oriented and higher scores indicate lower stress levels for the job control and social support items. In contrast, the other items are negatively oriented and higher scores indicate greater stress levels.

### 2.3. Measures of Depressive and Anxiety Symptoms

We evaluated depressive symptoms using the Japanese version of the PHQ-9, which has been validated [38,39]. The scale includes nine items regarding the frequency of problems bothering participants in the past two weeks, and each item is rated from 0 (not at all) to 3 (almost every day). The total scores range between 0 and 27 and the higher scores indicate a higher severity of depressive symptoms. Previous studies used a score ≥5 to indicate the presence of elevated depressive symptoms [27,28,29,38,40]; thus, we used this cut-off point to divide participants into a Depressive group (DEP) and non-depressive group (non-DEP). We evaluated anxiety symptoms using the Generalized Anxiety Disorder Questionnaire 7-item scale (GAD-7) [41], with each question rated from 0 (not at all) to 3 (almost every day). The total scores range from 0 to 21 and higher scores indicate a higher severity of anxiety symptoms. In previous studies, a score ≥5 was used to indicate the presence of anxiety symptoms [27,28,29,40,41]; thus, we used this cut-off point to divide participants into an anxiety group (ANX) and a non-anxiety group (non-ANX).

### 2.4. Demographic, Work-Related, and COVID-19-Related Variables

The participants also reported their demographic information, including age, gender, marital status, number of children, education, family income, and alcohol consumption. We also collected information on work-related variables: occupation, type of employment, position classification, work pattern, frequency of working at home, service years, and overtime hours per month. Further, we collected demographic and work-related information that was associated with COVID-19: living in COVID-19 special precautions areas (i.e., Hokkaido, Tokyo, Kanagawa, Saitama, Chiba, Aichi, Osaka, Hyogo, Kyoto, Fukuoka, and Okinawa) or not; the incidence of familiar persons infected by COVID-19; the incidence of contact with a COVID-19 patient; physical distance to coworkers: ≥1.5m or not, physical distance to clients: ≥1.5m or not; and anxiety level for COVID-19. Anxiety and fear for COVID-19 were measured by asking, “Are you worried about COVID-19?” The responses were scored on a 7-point Likert-type scale (ranging from 1 = Not at all to 7 = Feel strongly). The higher the score, the greater the level of COVID-19 anxiety.

### 2.5. Statistical Analyses

Using the cut-off point, we divided the participants into a DEP and non-DEP group and an ANX and non-ANX group, respectively. We defined the inclusion criteria for the DEP and ANX groups as dependent variables, and we defined the demographic variables, work-related variables, COVID-19-related variables, and the six GJSQ subscales as independent variables. Univariate logistic regression analyses were performed to estimate the adjusted odds ratios (AORs) of the demographic variables, work-related variables, COVID-19-related variables, and the six GJSQ subscales for the DEP and ANX groups. According to the tertile scores, the GJSQ subscales were subdivided into low, moderate, and high categories. Subsequently, the AORs for belonging to the DEP and ANX groups were estimated by multivariate logistic regression analyses and adjusted for all the demographic variables, work-related variables, COVID-19-related variables, and the six subdivided GJSQ subscales. Statistical significance was set at *p* < 0.05. Statistical data analyses were performed by SPSS version 26.0 software (SPSS Inc., Chicago, IL, USA).

## 3. Results

Table 1 shows the demographic, work-related, and COVID-19-related characteristics of the participants. Six-hundred twenty-one (77%) participants were male, with a mean age of 46.6 years (SD = 10.5). Regarding the work-related variables, 659 (81.7%) participants were regular workers, 688 (85.3%) daytime workers, 440 (54.5%) non-managers, and 578 (71.6%) reported to their workplace daily. The average service years were 13.9 years (SD = 11.2) and the overtime hours per month were 16.2 h (SD = 20.5). Regarding the COVID-19-related characteristics, 530 (65.7%) participants lived in special precautions areas due to COVID-19, 68 (8.4%) had at least one family member or coworker who was infected with COVID-19, and 58 (7.2%) had contact with at least one COVID-19 patient. Five hundred twenty-six (65.2%) participants worked in a setting without a physical distance of more than 1.5 m among coworkers, and 691 (85.6%) worked without a physical distance of more than 1.5 m to clients. Table 2 shows the GJSQ, PHQ-9, and GAD-7 scores of 807 non-HCWs. The PHQ-9 scores were 5.2 (SD = 5.0) and GAD-7 scores were 3.8 (SD = 4.5). The prevalence of depressive and anxiety symptoms was 43.4% and 31.8%, respectively.

Table 3 and Table 4 show the results of the logistic regression analysis. The AORs were calculated using the demographic variables, work-related variables, COVID-19-related variables, and each of the six subdivided GJSQ subscales as independent variables, with the DEP and ANX groups as dependent variables. When entering the independent variables in a multivariate logistic regression analysis, the depressive symptoms were associated with a family income of more than 12 million yen (approximately USD 110 thousand; AOR = 0.45, 95% CI = 0.20–0.99); being an executive (AOR = 0.22, 95% CI = 0.07–0.75); having contact with a COVID-19 patient (AOR = 3.14, 95% CI = 1.60–6.18); “job future ambiguity” for the participants with a moderate (AOR = 1.83, 95% CI = 1.17–2.87) and high (AOR = 2.17, 95% CI = 1.39–3.39) level of stress; “variance in workload” for participants with a high level of stress (AOR = 2.20, 95% CI = 1.36–3.55); and “social support from coworkers” for participants with a high level of stress (AOR = 0.36, 95% CI = 0.21–0.62). Anxiety symptoms were associated with having contact with a COVID-19 patient (AOR = 2.85, 95% CI = 1.48–5.49); “job future ambiguity” for participants with a moderate (AOR = 1.89, 95% CI = 1.17–3.06) and high (AOR = 2.05, 95% CI = 1.27–3.30) level of stress; “variance in workload” for participants with a high level of stress (AOR = 2.05, 95% CI = 1.25–3.37); and “social support from coworkers” for participants with a high level of stress (AOR = 0.37, 95% CI = 0.21–0.65). All the results of the logistic regression analysis are presented in the Table 3 and Table 4.

## 4. Discussion

The results indicate a relationship between the variance in workload, job future ambiguity, social support from coworkers, having contact with a COVID-19 patient, and depressive and anxiety symptoms, and a relationship between socioeconomic status and depressive symptoms. However, there were no relationships between the quantitative workload, job control, depressive, and anxiety symptoms. To our knowledge, this is the first study to identify the relationships between occupational stress and change in the work environment due to COVID-19 and depressive and anxiety symptoms among non-HCWs during the COVID-19 pandemic. 

Kuzman et al. proposed eight basic principles for the organization of mental health care. They emphasized that there should be no substantial differences in the provision of health care for COVID-19 between persons with pre-existing mental health disorders and those without previous disorders [42]. McDaid D stated that the economic recovery in Europe depends on the physical and mental health of its citizens; the support for mental health recovery needs to be accurately portrayed as a positive investment that will benefit society rather than a cost to be minimized [43]. Non-HCWs are both persons with and those without pre-existing mental health disorders. Therefore, careful attention should be paid to them for the benefit of society.

The prevalence of depressive and anxiety symptoms was 43.4% and 31.8%, respectively, in this study. Previous studies conducted with the general population during the COVID-19 pandemic, using cut-off scores ≥5 on the PHQ-9 and the GAD-7, reported that the prevalence of depressive symptoms (i.e., PHQ-9) was 52.5% in the United States [44] and 27.9% [27] and 43.7% [29] in China, and the prevalence of anxiety symptoms (i.e., GAD-7) was 31.6% [27] and 37.4% [29] in China. This study was conducted during the third wave of the COVID-19 pandemic in Japan when the number of new COVID-19 cases, new COVID-19 serious cases, and hospitalized COVID-19 patients increased more rapidly than in the first and second waves, and Japan’s second state of emergency was enforced. These conditions were the reasons for the relatively high prevalence of depressive and anxiety symptoms among non-HCWs in this study.

This study identified the relationship between having contact with a COVID-19 patient and depressive and anxiety symptoms. Being infected by either confirmed or suspected COVID-19, having any family members or friends infected by COVID-19, and having occupational exposure risks to patients infected with COVID-19 were found to be risk factors for elevated depressive and anxiety symptoms among the general population in China [27]. Another study of 123,768 factory workers during the pandemic in China, reported that having COVID-19 confirmed cases in the community and having COVID-19 confirmed friends were associated with an increased risk of depressive and anxiety symptoms [19]. Similarly, an increased risk of depressive symptoms during the COVID-19 pandemic was associated with greater exposure to stress (e.g., losing a job, the death of someone close owing to COVID-19, and having financial problems) in the United States [44]. The Centers for Disease Control and Prevention (CDC) indicated that the “Concern about the risk of being exposed to the virus at work” was a potential common work-related factor [45]. The results of this study were consistent with previous studies. Reducing contact with COVID-19 patients in the workplace appears to be a factor that may protect against depressive and anxiety symptoms among non-HCWs.

The relationships between job future ambiguity, the variance in workload, support from coworkers, and depressive and anxiety symptoms during the COVID-19 pandemic were also demonstrated in this study. However, there were no relationships between the quantitative workload, job control, and social support from supervisors and depressive and anxiety symptoms. Before the COVID-19 pandemic, many longitudinal studies and meta-analyses had demonstrated that a higher quantitative workload, lower job control, lower social support, and a higher job strain were associated with the risk of depressive symptoms [20,21,22,23]. These occupational stress were primarily viewed as risk factors for mental health problems in the workplace. Job future ambiguity refers to any uncertainty regarding a promotion, skill development, or increase in responsibilities that an individual can experience at work. As for job future ambiguity, before the COVID-19 pandemic, the relationships between depressive symptoms and job future ambiguity were reported among HCWs in public hospitals in Qatar [46], and we demonstrated that anxious temperament predicted a higher level of job future ambiguity among Japanese civil servants [47]. The variance in workload refers to the extent of a marked increase in workload, the amount of concentration required on work tasks, and the speed required to complete work tasks. As for the variance in workload, before the COVID-19 pandemic, previous research indicated that variance in workload was associated with depressive symptoms among Japanese firefighters [37] and heavy drinking among female Japanese teachers [36]. However, little research on job future ambiguity and variance in the workload as occupational stress has been conducted. No studies have focused on the relationship between variance in workload and depressive and anxiety symptoms, and a few studies have focused on the relationship between job future ambiguity and depressive and anxiety symptoms during the COVID-19 pandemic. A study in Serbia among different categories of employees, demonstrated that job uncertainty and the fear of COVID-19 related to work-related distress [48]. In the U.S.A., another study among white-collar employees indicated that job insecurity has a substantial impact on depressive and anxiety symptoms, [49]. Moreover, few studies have mentioned non-HCWs’ occupational stress during the COVID-19 pandemic, although the CDC has indicated potential common work-related factors that can increase stress during the COVID-19 pandemic; these include the “Uncertainty about the future of your workplace and/or employment,” “Managing a different workload,” “Lack of access to the tools and equipment needed to perform your job,” “Learning new communication tools and dealing with technical difficulties,” and “Adapting to a different workspace and/or work schedule” [45]. These factors identified by the CDC support the results of this study.

The COVID-19 pandemic lasted globally for more than one year. It has led to drastic workstyle changes in the workplace, economic deterioration, dismissal, and feelings of job insecurity or ambiguity, leading to an increase in “job future ambiguity.” Further, the unexpected and sudden change in the work environment, such as coworkers’ absence due to a SARS-CoV-2 infection, having close contact with a COVID-19 patient (e.g., self, coworker, and family), and a change of work policy and workstyle (e.g., telework and remote working) lead to an increase in “variance in workload.” These two occupational stresses are expected to have a large influence on the mild depressive symptoms and anxiety of non-HCWs. Conversely, the support from coworkers appears to serve a protective role against depressive and anxiety symptoms of non-HCWs, although maintaining frequent in-person social contact has been difficult during the pandemic. Sufficient communication with coworkers was a source of social support [50] to reduce emotional exhaustion [51] before the pandemic. A longitudinal study during the pandemic among software professionals living in the U.S.A. and Europe indicated that promoting effective communication among coworkers can help to maintain a better mental health status [52]. Therefore, promoting indirect contact with coworkers using online communication tools and supporting each other will be essential during the COVID-19 pandemic.

The effect of variance in workload, job future ambiguity, and contact with COVID-19 patients on depressive and anxiety symptoms might be greater for non-HCWs during the COVID-19 pandemic rather than before the pandemic, and the effect of occupational stress on depressive and anxiety symptoms during the COVID-19 pandemic might differ from that before the pandemic.

For companies and workplaces, some countermeasures for employees’ occupational stress are needed during the COVID-19 pandemic. First, clarifying the job and company’s prospects, even if the prospects are good or bad, are likely to reduce “job future ambiguity.” Non-HCWs strongly fear being out of business and becoming downsized; therefore, employment stability is likely the greatest protective factor against anxiety and depressive symptoms. Second, securing adequate human resources and reducing the workload is critical for reducing the occupational stress related to “variance in workload.” Third, creating a work environment that supports communication among coworkers and strengthens social support among coworkers is important. Enhanced social support by focusing on changes in workers’ performance (reduced work efficiency) and interactions (the deterioration of relationships with colleagues and superiors) in the workplace are necessary for the early detection of major depressive disorder [53]. Fourth, taking all available COVID-19 precautions, such as assessing job hazards for the feasibility of engineering controls, ensuring ventilation and water systems operate properly, altering workspaces to maintain physical distancing, temperature and symptom screening, encouraging sick workers to report symptoms, encouraging physical distancing and wearing masks in the workplace, and using technology to promote physical distancing [54], are essential for protecting employees from SARS-CoV-2 infection.

Non-HCWs should become better at recognizing their symptoms of stress (e.g., feeling irritable, angry, nervous, anxious, tired, overwhelmed, burned out, depressed, or experiencing sleeplessness) [45], consulting reliable persons (e.g., family, friends, coworkers, supervisors, or counselors), and having various coping skills for depressive and anxiety symptoms. For example, cognitive-behavioral therapy [55,56] and mindfulness [57] are effective for employees’ prevention of depressive and anxiety symptoms. Similarly, Morita therapy, an indigenous Japanese therapy from Masatake Morita [58,59,60], has been proven as effective in reducing depressive and anxiety symptoms.

There are several limitations to the present study. First, it may be difficult to generalize our findings to different countries and cultures as only Japanese non-HCWs were surveyed and the infection status, infection control measures, and type of lockdown implemented to control the COVID-19 pandemic differ among countries. Further, the number of participants was small, the data were obtained from web-based sources, and the representativeness of the study sample is unknown. Second, occupational stress and depressive and anxiety symptoms were assessed using self-report; thus, the response bias may have influenced the results, and some misclassification might exist. Third, a cross-sectional design was used; therefore, our findings cannot examine the causal relationship between occupational stress and depressive and anxiety symptoms, nor can the directionality of any such relationship be established. A cohort or longitudinal design is necessary to examine this causal relationship in-depth among non-HCWs and would be beneficial in the future. Fourth, we have no data on the same workers without the effect of COVID-19, and the number of HCWs in this study was too small. Therefore, there is no other group of workers for comparison. Fifth, there is uneven gender distribution as the study sample comprised of 77% males. Sixth, anxiety and fear for COVID-19 were measured in this study by asking one question, “Are you worried about COVID-19?”. This is an original tool created by us and its reliability and validity have not been assessed.

## 5. Conclusions

We identified two significant relationships in this study: a relationship between variance in workload, job future ambiguity, social support from coworkers, contact with COVID-19 patients, and depressive and anxiety symptoms, and a relationship between the socioeconomic status and depressive symptoms among Japanese non-HCWs during the COVID-19 pandemic. In addition, the effect of occupational stress on depressive and anxiety symptoms during the COVID-19 pandemic differed from the effects reported before the pandemic.

Paying attention to job future ambiguity, variance in workload at the workplace and individual perspectives, promoting contact among coworkers using online communication tools, and reducing contact with COVID-19 patients are likely to be useful in decreasing the depressive and anxiety symptoms of non-HCWs.

## Figures and Tables

**Table 1 ijerph-19-00983-t001:** Participants’ socio-demographic characteristics and work environment (N = 807).

Age (Years)	n (%)
20–29	55 (6.8)
30–39	153 (19.0)
40–49	259 (32.1)
50–59	244 (30.2)
≥60	96 (11.9)
Gender	
Male	621 (77)
Female	186 (23)
Marital status	
Married	523 (64.8)
Single	284 (35.2)
Child(ren)	
None	346 (42.9)
≥1	461 (57.1)
Education (years)	
≤12	183 (22.7)
≥13	624 (77.3)
Family income (million yen)	
<4	182 (22.6)
4–8	374 (46.3)
8–12	187 (23.2)
>12	64 (7.9)
Occupation	
Clerical worker	238 (29.5)
Technical worker	206 (25.5)
Workers (not clerical and technical)	204 (25.3)
Civil servants	56 (6.9)
Executives	28 (3.5)
Self-employment	75 (9.3)
Type of employment	
Regular	659 (81.7)
Temporary	148 (18.3)
Position classification	
Non-manager	440 (54.5)
Manager	367 (45.5)
Work pattern	
Daytime	688 (85.3)
Shift	119 (14.7)
Frequency of working at home	
None (every day at workplace)	578 (71.6)
≤Twice per week	103 (12.8)
>3 times per week	126 (15.6)
Service years	
Overtime (hours/month)	
Alcohol consumption	
None	215 (26.7)
Once per month	121 (15)
<1 per week	131 (16.2)
2–3 times per week	97 (12)
>4 times per week	243 (30.1)
Special precautions area due to COVID-19	
Yes	530 (65.7)
No	277 (34.3)
Familiar person infected COVID-19 (family, coworker)	
Positive	68 (8.4)
None	739 (91.6)
Contact with COVID-19 patient	
Positive	58 (7.2)
None	749 (92.8)
Physical distance to a coworker: ≥1.5m	
Positive	281 (34.8)
None	526 (65.2)
Physical distance to a client: ≥1.5m	
Positive	116 (14.4)
None	691 (85.6)

**Table 2 ijerph-19-00983-t002:** Participants’ occupational stress, depressive, and anxiety symptoms (N = 807).

	Range	Mean (SD)
Anxiety for COVID-19	1–7	4.6 (1.8)
GJSQ scores		
Quantitative workload	11–55	34.4 (6.4)
Job control	16–80	47.8 (10.7)
Job future ambiguity	4–20	17.4 (5.4)
Variance in workload	3–15	9.3 (2.5)
Supervisors’ support	4–20	12.7 (3.9)
Coworkers’ support	4–20	13.3 (3.7)
PHQ-9	0–27	5.2 (5.0)
GAD-7	0–21	3.8 (4.5)

Abbreviations: GJSQ: Generic Job Stress Questionnaire; PHQ: Patient Health Questionnaire; GAD-7: Generalized Anxiety Disorder 7-item Scale; and SD: Standard deviation.

**Table 3 ijerph-19-00983-t003:** Multiple logistic regression analysis of risk factors related to the DEP group, compared with the non-DEP group.

	Univariate Model			Adjusted Model *		
	AOR	(95% CI)	*p*	AOR	(95% CI)	*p*
Family income (million yen)						
<400	Ref			Ref		
400–800	0.76	(0.53–1.08)	0.121	0.87	(0.57–1.32)	0.520
800–1200	0.55	(0.36–0.83)	<0.01	0.65	(0.38–1.12)	0.119
>1200	0.31	(0.16–0.58)	<0.001	0.45	(0.20–0.99)	0.040
Occupation						
Clerical worker	Ref			Ref		
Technical worker	0.98	(0.68–1.43)	0.928	1.22	(0.78–1.93)	0.390
Workers (not clerical and technical)	1.19	(0.82–1.74)	0.354	1.33	(0.84–2.08)	0.220
Civil servants	0.70	(0.39–1.29)	0.254	0.88	(0.44–1.79)	0.73
Executives	0.21	(0.07–0.63)	<0.01	0.22	(0.07–0.75)	0.015
Self-employment	0.94	(0.56–1.59)	0.825	0.70	(0.35–1.39)	0.310
Contact with a COVID-19 patient						
None	Ref			Ref		
Positive	2.66	(1.52–4.66)	<0.01	3.14	(1.60–6.18)	<0.001
Occupational stress						
Quantitative workload						
Low	Ref			Ref		
Moderate	1.40	(0.98–2.00)	0.067	1.18	(0.75–1.85)	0.470
High	1.47	(1.03–2.09)	<0.05	1.11	(0.68–1.83)	0.670
Job control						
High	Ref			Ref		
Moderate	1.54	(1.09–2.17)	<0.05	1.14	(0.74–1.77)	0.550
Low	1.50	(1.07–2.10)	<0.05	1.31	(0.87–1.98)	0.200
Job future ambiguity						
Low	Ref			Ref		
Moderate	2.11	(1.44–3.09)	<0.001	1.83	(1.17–2.87)	<0.01
High	2.86	(1.96–4.17)	<0.001	2.17	(1.39–3.39)	<0.001
Variance in workload						
Low	Ref			Ref		
Moderate	1.18	(0.81–1.71)	0.401	1.26	(0.79–2.02)	0.340
High	1.79	(1.28–2.52)	<0.01	2.20	(1.36–3.55)	<0.001
Social support from supervisor						
Low	Ref			Ref		
Moderate	0.49	(0.35–0.69)	<0.001	0.67	(0.44–1.04)	0.080
High	0.30	(0.21–0.43)	<0.001	0.64	(0.38–1.07)	0.090
Social support from coworker						
Low	Ref			Ref		
Moderate	0.51	(0.36–0.74)	<0.001	0.65	(0.41–1.02)	0.060
High	0.26	(0.18–0.38)	<0.001	0.36	(0.21–0.62)	<0.001

*: Adjusted for all the listed variables. Abbreviations: CI = Confidence interval; AOR = Adjusted odds ratio; and DEP = Depressive.

**Table 4 ijerph-19-00983-t004:** Multiple logistic regression analysis of the risk factors related to the ANX group, compared with the non-ANX group.

	Univariate Model			Adjusted Model *		
	AOR	(95% CI)	*p*	AOR	(95% CI)	*p*
Contact with COVID-19 patient						
None	Ref			Ref		
Positive	2.46	(1.44–4.22)	<0.001	2.85	(1.48–5.49)	<0.01
Occupational stress						
Quantitative workload						
Low	Ref			Ref		
Moderate	1.29	(0.88–1.90)	0.192	1.12	0.69–1.79	0.650
High	1.41	(0.97–2.06)	0.074	1.04	0.62–1.73	0.900
Job control						
High	Ref			Ref		
Moderate	1.30	(0.91–1.87)	0.155	1.11	0.70–1.74	0.660
Low	1.01	(0.71–1.45)	0.940	0.83	0.54–1.29	0.410
Job future ambiguity						
Low	Ref			Ref		
Moderate	1.96	(1.29–2.97)	<0.01	1.89	1.17–3.06	<0.01
High	2.49	(1.65–3.75)	<0.001	2.05	1.27–3.30	<0.01
Variance in workload						
Low	Ref			Ref		
Moderate	1.15	(0.77-1.74)	0.494	1.30	0.78–2.16	0.310
High	1.76	(1.22-2.53)	<0.01	2.05	1.25–3.37	<0.01
Social support from supervisor						
Low	Ref			Ref		
Moderate	0.54	(0.38–0.76)	<0.01	0.68	0.44–1.05	0.080
High	0.32	(0.22–0.47)	<0.001	0.70	0.40–1.20	0.190
Social support from coworker						
Low	Ref			Ref		
Moderate	0.57	(0.39–0.82)	<0.01	0.78	0.49–1.22	0.280
High	0.26	(0.18–0.39)	<0.001	0.37	0.21–0.65	<0.001

*: Adjusted for all the listed variables. Abbreviations: CI = Confidence interval; AOR = Adjusted odds ratio; and ANX = Anxiety.

## Data Availability

All the data generated or analyzed during this study are included in this published article.

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
