# Peer review of "Relationships between Occupational Stress, Change in Work Environment during the COVID-19 Pandemic, and Depressive and Anxiety Symptoms among Non-Healthcare Workers in Japan: A Cross-Sectional Study"

_ijerph, 2022, doi:10.3390/ijerph19020983_

Round 1
Reviewer 1 Report
The study is valuable and topical because it deals with the mental health of workers during the COVID-19 pandemic. This study aimed to clarify the effect of occupational stress and change in the occupational environment on non-healthcare workers’ mental health during the third wave of the COVID-19 pandemic in Japan. Occupational stress, depressive and anxiety symptoms were assessed by standardized and validated questionnaires.
The article is written clearly in good English. Up-to-date references are included. The statistical analysis is appropriate and the statistical program up to date.
However, there are some shortcomings and limitations that need to be revised.
- The title is too long.
- There is no other group of workers to compare.
- Ethical committee approval is not mentioned in the methods section of the article.
- The title of Table 1 is not explained properly, what kind of characteristics are explained in this table?
- Table 1 has too much information and too much space at the beginning, two tables should be made from that data.
- There is a typo in Table 1 – „Psysical distance to coworker“, should be „Physical distance to a coworker“.
- There is not 100% total in some Table 1 variable categories.
- There were no explanations about how was multiple logistic regression performed and „adjusted odds ratios“ were assessed, not „odds ratios“.
- Tables 2 and 3 – it is not clear which groups are compared in the logistic regression analysis.
- Please, describe the category of non-healthcare workers, what kind of workers are included? Blue-collar or white-collar workers, teachers, lawyers, or craftsmen.
- The problem could be also uneven gender distribution with 77 % males (not mentioned in the limitations of the study).
Reviewer 2 Report
Aim of the study is to evaluate the effect of occupational stress and changes in the work environment on non-Health Care Workers’ (HCWs) mental health during the third wave of the 12 COVID-19 pandemic in Japan.Various Questionannaires have been used for a web-based survey in 807 non- HCWs.The Authors report a relationship between variance in workload, job future ambiguity, social 21 support from coworkers, having contact with COVID-19 patients, and depressive and anxiety 22 symptoms.They suggest solutions for decreasing the depressive and anxiety symptoms among non-HCWs.This study is very interesting overall for the few data in the licterature about the effect of Covid on HCWs.The used Questionnaries are appropriate although there are limits in web administered tools in this field.Could been interesting a follow-up of this study comparing data from Health Care Workers and non- Health Care Workers’ to evaluate the differences in these categories.
This study is well conducted and can be useful for future evaluations and researches
Reviewer 3 Report
This is a well-written paper with many interesting findings. The empirical study was conducted in a good design but I couldn't find the reference to the ethical permission.
However, the conclusions show a perspective, as the authors also mentioned they have a limited ecological validity. The authors may search for articles on the topic for European and American results.
By the 3. section (results) data and text are at the beginning (eg. the sample description) a bit redundant but in the other parts of the sections, some text as an interpretation of long tables would be helpful, or authors may show only the significant and important results. English proofreading is also needed and authors should check the text, citations, name, and number of tables and correct the missing figure that they mentioned.
Authors refer many times to Figure 1 in the text but there are no figures in the article, please check and correct it.
Although the last section could be improved the topic is important, the empirical study is well designed and researchers had a sample big enough to make drive the conclusions. The structure of the paper is correct and the findings are interesting.
Round 2
Reviewer 1 Report
The article has improved a lot after all those corrections. I still have some comments:
- I think adjusted odds ratios are in the adjusted model, the univariate model has just simple odds ratios, lines 191 -202 - there are only ORs mentioned, but should be AORs
- Table 1 looks strange to me, with a lot of empty space for means and SDs - those three means and SD could be just mentioned in the text, without making the table strange, with a lot of empty space.
- Lines 343-351, in Conclusion, could fit much better to Discussion.
